

# Impacts of a decadal drainage disturbance on surface–atmosphere fluxes of carbon dioxide in a permafrost ecosystem

F. Kittler[1], I. Burjack[1], C. A. R. Corradi[2], M. Heimann[1,3], O. Kolle[1], L. Merbold[4], N. Zimov[5], S. A. Zimov[5] , M. Göckede[1]

[1] Biogeochemical Systems, Max Planck Institute for Biogeochemistry, Jena, Germany
[2] University of Tuscia of Viterbo, Viterbo, Italy
[3] Division of Atmospheric Sciences, Department of Physics, University of Helsinki, Finland
[4] ETH Zurich, Department of Environmental Systems Science, Institute of Agricultural Sciences, Zurich, Switzerland
[5] North-East Science Station, Pacific Institute for Geography, Far-Eastern Branch of Russian Academy of Science, Chersky, Republic of Sakha (Yakutia), Russia

*Correspondence to*: F. Kittler (fkittler@bgc-jena.mpg.de)

**Abstract.** Hydrologic conditions are a major controlling factor for carbon exchange processes in high-latitude ecosystems. The presence or absence of water-logged conditions can lead to significant shifts in ecosystem structure and carbon cycle processes. In this study, we compared growing season $CO_2$ fluxes of a wet tussock tundra ecosystem from an area affected by decadal drainage and an undisturbed area on the Kolyma floodplain in northeastern Siberia. For this comparison we found the $CO_2$ uptake to be systematically reduced within the drained area, with a minor increase in photosynthetic uptake due to a higher abundance of shrubs outweighed by a more pronounced increase in respiration due to warmer near-surface soil layers. Carbon dioxide exchange with the atmosphere over this disturbed part of the tundra has rebounded from the strong reduction of fluxes immediately following the drainage disturbance in 2005. This indicates that the local permafrost ecosystem is capable of adapting to significantly different hydrologic conditions without losing its capacity to act as a net sink for $CO_2$ over the growing season. The comparison of undisturbed $CO_2$ flux rates from 2013–2015 to the period of 2002–2004 indicates that overall $CO_2$ exchange with the atmosphere was intensified. Analyzing trends in component fluxes (ecosystem respiration and gross primary production) over the past decade, we found that net changes in $CO_2$ exchange fluxes are dominated by a major increase in photosynthetic uptake, resulting in a stronger $CO_2$ sink in 2013–2015. Application of a MODIS-based classification scheme to separate the growing season into four sub-seasons improved the interpretation of interannual variability, and helps in illustrating the systematic shifts in $CO_2$ uptake patterns that have occurred in this ecosystem over the past 10 years.

## 1 Introduction

Northern high latitude permafrost landscapes contain about 1700 Pg of organic carbon in the upper 3 m of the soil (Tarnocai et al., 2009), equaling approximately 50 % of the global belowground carbon storage. This vast amount of organic carbon has been accumulated due to the slow decomposition rates of organic matter under low temperatures and anoxic





conditions during the last two glacial cycles (Zimov et al., 2009). With the sustainability of these huge carbon pools being dependent on future climate conditions (Kaufman et al., 2009; Kirschbaum, 1995; Serreze et al., 2000), there is a high potential for global feedback if the permafrost carbon reservoir is destabilized under future climate conditions (Beer, 2008).

One potential outcome could be that the soil carbon previously locked away in permafrost deposits could be released into the atmosphere as $CO_2$ or $CH_4$ as a consequence of changes in soil temperature and moisture conditions (Schuur et al., 2008). However, model simulations show a wide range of permafrost responses to climate change (Koven et al., 2011; Schaefer et al., 2011). Empirical studies on the sensitivity of permafrost to environmental drivers are urgently needed to corroborate comprehensive Earth System models. Here we report on the effects of a decadal drainage experiment on the $CO_2$ exchange

in a typical northern Siberian tundra ecosystem.

Over the last few decades observations in the Arctic have shown that, as a result of global warming, temperatures have risen faster than those observed over the Northern Hemisphere as a whole (ACIA, 2005; SWIPA, 2011). This tendency, primarily driven by ice-albedo effects (Curry et al., 1995), is expected to continue, and climate models project a strong future high-latitude warming (IPCC, 2013). In permafrost ecosystems, higher temperatures will lead to increased active layer

thickness, and thus favor substrate availability for microbial decomposition and enhanced release of greenhouse gases (GHGs) to the atmosphere (Schuur et al., 2008). At the same time, warmer conditions may lead to prolonged vegetation periods with increased photosynthetic uptake, while higher soil temperatures may stimulate $CO_2$ production through enhanced soil respiration (Flanagan and Syed, 2011). Ice wedge degradation (Jorgenson et al., 2006; Lawrence et al., 2008; Payette et al., 2004) is also expected, which has the potential to dramatically alter the geomorphology and hydrology of

permafrost landscapes (Liljedahl et al., 2016). Through this process, a relatively homogeneous area could be transformed into a network of small water channels that drain water away from the remaining patches of soil surface. The resulting changes in the spatiotemporal patterns of soil water availability could potentially trigger significant shifts in ecosystem characteristics, such as soil thermal regime or vegetation and microbial community composition. These changes, in turn, would alter carbon cycle processes; for example, changing aerobic conditions in the top soil layers would change microbial

decomposition from methanotrophic to oxidative processes (Merbold et al., 2009; Zona et al., 2009; 2010). Such changes would promote $CO_2$ efflux due to higher soil respiration rates (Flanagan and Syed, 2011; Olivas et al., 2010).

The complexity of potential positive and negative feedback loops between climate change and carbon cycling in the Arctic leads to considerable uncertainties concerning the effects of long-term shifts in the carbon budget of an Arctic permafrost ecosystem caused by changing environmental conditions. Most studies to date on the net effect of such a

disturbance on biogeochemical processes in the Arctic have focused on short-term perturbations (Bret-Harte et al., 2013; Merbold et al., 2009); however, it is likely that the long-term effects of this disturbance may differ greatly from the short-term effects that immediately follow the disturbance event (Shaver et al., 1992). Studies addressing long-term (>10 years) manipulation effects for the high latitudes are rare. In one such study, Lamb et al. (2011) found that 16 years of a combined fertilization and warming experiment in Canada's high Arctic tundra showed limited effects on GHG fluxes and soil

chemistry or biochemistry, but strong increases in plant cover and height. A two-decade warming experiment in Alaskan



tundra revealed increased plant biomass and woody dominance, indirectly increased winter soil temperature, and increased net ecosystem carbon storage (Sistla et al., 2013). In wetting and fertilizing experiments in the open heath of Zackenberg, an area in northeastern Greenland, the soil water supply was increased for 14 years, and occasional fertilizer pulses were added, resulting in increased soil respiration rates during summer and decreased $CO_2$ efflux in autumn (Christiansen et al., 2012).

Results from a summer warming and wetting treatment in northeastern Greenland over 10 years indicated increased $CO_2$ uptake in response to altered environmental conditions; this increase was mostly cause by photosynthetic uptake of carbon (Lupascu et al., 2014). Due to the multiple and unclear results of available environmental manipulations, additional and particularly long-term experiments are crucially needed.

To improve the current understanding of links between carbon fluxes and long-term hydrologic disturbance in Arctic

permafrost ecosystems, a monitoring program was established near Chersky in northeastern Siberia in 2013 (68.75 °N, 161.33 °E). In this study, we present growing season $CO_2$ fluxes for two eddy-covariance towers running in parallel over a disturbed tundra ecosystem (i.e., one with a drainage ditch ring installed in 2004) and a reference tundra ecosystem, respectively. We directly compare $CO_2$ flux rates between both sites for three growing seasons (2013–2015) to evaluate the net effect of the long-term drainage on the carbon cycle. A comparison of disturbed and reference flux rates between historic

(2002–2005) and recent (2013–2015) datasets allows us to assess the combined effect of this disturbance and a changing climate. In both cases, we focus on the net $CO_2$ flux between ecosystem and atmosphere, as well as its major components fluxes, i.e., gross primary productivity (GPP) and ecosystem respiration (Reco). We also analyzed the seasonality of flux rates over the course of the growing season.

## 2 Methods and Datasets

### 2.1 Site description

The research site is located in a floodplain of the Kolyma river near the city of Chersky (68.75 °N, 161.33 °E), in the northeastern part of Siberia, Russia. Situated south of the Arctic tree line within the floodplain of the Kolyma river, this area is subject to annual flooding in spring, leading to a common vegetation structure that is characterized by sedges, shrubs, and sphagnum mosses (Merbold et al., 2009) and defined as a wet tussock tundra. The mineral soil of the permafrost and active

layer is influenced by the sediment transport of the river and is classified as silty loam (Corradi et al., 2005). The transition period between the snow-covered and snow-free seasons is strongly influenced by the flooding in late spring, which in most years leads to standing water on the site as a result of snowmelt and rising water levels in the nearby river. Development of the vegetation in the flooded sections is therefore usually delayed compared to dry areas where snow melt is followed by rapid greening. Continuous daylight and continental air masses (which are warm, dry, and from the S to SE) characterize the

summer (i.e., June–August). Overall, in 2014, the mean annual temperature was -10 °C, with daily means ranging from -49 °C in January to +22 °C in July. The mean annual precipitation was about 172 mm, with nearly 50 % of this precipitation




occurring during the summer. As a result of limited light and low temperatures, the growing season is mainly restricted to three months of the year, from mid-June to the end of August.

## 2.2 Disturbance experiment

The experimental concept of the disturbance is designed to modify the local water regime and soil moisture conditions. The manipulation consists of a 200 m diameter drainage ditch that connects to the nearby river through a channel of ~ 300 m length (Merbold et al., 2009). This drainage system was installed in Autumn 2004 and is still in place today, altering the water table depth and thus significantly modifying biogeochemical and biogeophysical site characteristics during the last decade. For the recent observation program, two observation sites were chosen: one within the drainage ring (and thus
covering the areas affected by lowered ground water levels) and another serving as a reference site (close by but far enough away to represent natural conditions unaffected by the disturbance). Since both sites had to be close enough to share a power supply, the two observation sites are ~ 600 m apart (Fig. 1).

## 2.3 Instrumentation setup

       To monitor ecosystem-scale $CO_2$ fluxes, two eddy-covariance systems were installed in July 2013. One tower (Tower 1,
68.61 °N and 161.34 °E) was placed within the drainage area (Fig. 1, blue shaded area), so that its footprint primarily covers areas affected by reduced ground water levels. The reference tower (Tower 2, 68.62 °N and 161.35 °E) represents natural conditions (Fig. 1). Data acquisition started 13 July 2013 for Tower 1 and 16 July 2013 for Tower 2.

       The ground under both towers has an elevation of approximately 6 m a.s.l., with an average vegetation height of 0.7 m in the surrounding area during peak growing season. Each tower is equipped with a heated sonic anemometer (uSonic-3
Scientific, Metek) that monitors the three-dimensional wind field and the sonic temperature at the top of the towers (at heights of 4.91 m and 5.11 m for Tower 1 and 2, respectively). During the first nine months of the experiment, the $CO_2$ and $H_2O$ flux densities were measured exclusively using open-path sensors (LI-7500, LI-COR Biosciences). These gas analyzers were placed next to the sonic anemometer, with a sensor separation of 0.38 m at both towers. In 2014, new closed-path gas analyzers (FGGA, Los Gatos Research Inc.) for monitoring GHG mixing ratios were added to the setup. These systems
consist of an inlet placed next to the sonic anemometer (vertical sensor separation: 0.30 m), a sampling line (heated and insulated Eaton Synflex decabon with 6.2 mm inner diameter and a length of 16.0 m and 12.8 m for Tower 1 and 2, respectively), and an external vacuum pump (N940, KNF). The flow rate of 13 L min$^{-1}$ (ambient pressure) translates into a replacement of sample air in the measurement cell at a frequency of ~ 2–2.5 Hz. Both open- and closed-path analyzers have been running in parallel at Tower 1 since installation, while the open-path analyzer on Tower 2 was disassembled in July
2014. An overview of the active times for the different gas analyzing systems is given in Fig. 2. Additional instrumentation to monitor environmental variables is listed in Table 1.





## 2.4 Data processing

The observed meteorological data (Table 1) are collected at 10 second intervals and stored on a data logger (CR3000, Campbell Scientific) as averages over 10 minute periods. The post-processing quality control scheme of the meteorological data includes a test for failure of the power supply, a test that checks the range of parameters within a certain time window, a flat lining test, and a spike test, as well as tests for identifying malfunctioning sensors and consistency limits. The final dataset was averaged to 30 minute intervals.

For the eddy-covariance towers, the data are collected at a rate of 20 Hz with analogue output for the gas analyzers. All analog signals are first transmitted to the sonic anemometer, where they are digitized and sent to the site computer jointly with the wind signal. Data acquisition is handled through the software package EDDYMEAS (Kolle and Rebmann, 2007) on a local computer at the field site. The processing of the high frequency eddy-covariance data is based on the software tool TK3 (Foken and Mauder, 2015), which implements the following major components: for both open- and closed-path gas analyzing systems, the flux processing includes (1) a 2D coordinate rotation of the wind field, (2) a cross-wind correction (Liu et al., 2001), and (3) a correction for loss in the high frequency range (Moore, 1986). For the closed-path systems, the greenhouse gas mixing ratios originally recorded as wet mole fractions are converted to dry mole fractions before processing by the TK3 software. To correct losses in the high frequency range that occur when gases are transferred to the closed-path analyzers through inlet tubes at flow rates much lower than the sampling frequency, cut-off frequencies of 2.5 Hz and 2 Hz for $CO_2$ and $H_2O$, respectively, were applied. For data collected by the open-path gas analyzers, the WPL correction (Webb et al., 1980) is also applied. To account for the self-heating of the open-path analyzer, a correction was applied following Burba et al.'s (2006) approach for inclined positions (~ 15 degree), using an optimized fractionation factor of 0.07 based on high quality data. This fractionation factor was subsequently used for the data post-processing of all open-path measurements used within the context of the present study.

Our post-processing quality control scheme is based on the quality flagging system initially proposed by Foken and Wichura (1996), which tests data for effects such as stationarity and well-developed turbulence. Afterwards, a set of additional tests is applied to flag implausible data points in the resulting flux time series. These tests include a check for absolute limits for net ecosystem exchange (NEE; 5 µmol m$^{-2}$ s$^{-1}$ < NEE < -15 µmol m$^{-2}$ s$^{-1}$), the open-path gas analyzing system status information (gain control maximum > 75), and overall errors in the log file recorded by the sonic anemometer, as well as a comparison of the absolute concentrations of $CO_2$ for the two towers for specific wind directions. This comparison of concentrations is performed to detect potential contamination by the exhaust plume of the generator (based on a more than 5 % difference criterion).

Gapfilling and flux partitioning are both based on the so-called marginal distribution sampling method (Reichstein et al., 2005), implemented through the R package REddyProc (https://r-forge.r-project.org/projects/reddyproc/). As there was a pronounced variability of environmental conditions during the growing season, the dataset was split into four different sub-seasons (see Sect. 2.6 for details), with individual gapfilling and flux partitioning data pools for each sub-season. The final



NEE time series used within the context of the presented study were merged from open-path (until April 2014) and closed-path (since April 2014) fluxes.

A linear regression with a moving window approach was used to analyze peak uptake with a window size of 14 days with an hourly shift chosen to account for short-term variabilities. Maximum negative slopes are identified as maximum uptake rates while the center of the window represents the day of year (DOY) with the maximum uptake.

**2.5 Historic dataset**

Eddy-covariance measurements were conducted during the 2002–2005 growing seasons in the context of the TCOS Siberia project (Corradi et al., 2005; Merbold et al., 2009, Fig. 2), whose flux measurements until the end of October 2004 represent an undisturbed tussock tundra (Merbold et al., 2009). After the installation of the drainage channel in Autumn 2004, flux measurements in the subsequent year indicated that this modification in the local water regime resulted in no
significant change to the seasonal budgets of net carbon dioxide fluxes (Merbold et al., 2009). However, these trends were based on a single year of observations only, and thus did not reflect the long-term effects of such a disturbance.

To facilitate a direct comparison between recent (2013–2015) and historic (2002–2005) observations, the raw datasets from the period of 2002–2005 were re-processed using the same data processing and quality assessment procedure outlined in Sect. 2.4. This re-processing caused minor changes in the flux rates as compared to the values reported in Corradi et al.
(2005) and Merbold et al. (2009), due to changed assumptions and inputs when using a different flux processing software. However, general patterns in interannual variability of fluxes were not affected. For the data collected in 2004, a parallel run of open- and closed-path gas analyzers resulted in systematic differences in the seasonal dynamics and final budgets (see also Merbold et al., 2009) for $CO_2$ and $H_2O$ fluxes. These differences remained ($R^2 = 0.75$) even after re-processing and applying additional corrections that have since become available (Burba et al., 2006), as well as analyzer-specific
corrections. Since the seasonal patterns in the open-path data displayed implausible patterns (e.g., missing release at the beginning of the growing season) we decided to use only the closed-path data for 2004 in this study.

**2.6 Remote sensing and additional data sources**

We used MODIS NDVI (MOD09GQ, MOD13Q1) with both daily and 16-day temporal resolution to analyze the interannual variability in phenology at the observation site. Raw data for the 250 m daily NDVI products were filtered for
outliers based on the 16-days composite NDVI time series, and smoothed using a moving average approach (window size = 10 days). Based on the GreenBrown R package (Forkel et al., 2013), the following phenological key dates were derived from trends and patterns in the resulting seasonal course of NDVI: start of season (SOS), peak of season (POS), and end of season (EOS). These key dates allow the growing season to be separated into four distinct sub-seasons: (1) pre-season (until SOS), (2) early-season (SOS to POS), (3) late-season (POS to EOS) and (4) post-season (after EOS). To mark the beginning of the
growing season (start of the pre-season), the first date when the daily mean temperature remained above four degrees centigrade for four consecutive days was used. The end of the growing season (end of the post-season) was set at when there



was a MODIS fractional snow cover > 50 %. Mean values for SOS, POS, and EOS, as well as year-to-year variability of these key dates within the period 2000–2015, are provided in Fig. 3.

Our webcam (CC5MPXWD, Campbell Scientific) provides an additional data source for vegetation characterization.
Pictures were taken daily at noon over the course of the growing season. Fig. 4 provides an overview of the vegetation dynamics from pre-season (Fig. 4a) towards post-season (Fig. 4d).

# 3 Results

## 3.1 Effects of drainage disturbance on recent carbon fluxes

Net ecosystem exchange and component fluxes varied considerably between 2013 and 2015 (Fig. 5 and Fig. 6).
Splitting the time series into sub-seasons allows for an in-depth analysis of the changing impact of the drainage effects on the $CO_2$ flux over the course of the growing season:

1.  Pre-season: The pre-season was characterized by a rapid increase in net radiation and rising air temperatures (see Table A1). In the weeks directly following snowmelt, cumulative NEE (Fig. 5a) shows a net loss of $CO_2$ from the system, while photosynthetic uptake is still weak. Both the drained and reference study sites are continuous sources
for $CO_2$, with net releases of ~ 19 (± 4) $gCO_2$-C $m^{-2}$ over the course of 20–30 days. The release of $CO_2$ in this period was more pronounced for the drained site than for the reference site (Fig. 6a).

2.  Early season: With the onset of strong photosynthetic uptake linked to rapidly evolving plant biomass (Fig. 4b), both the timing and initial rates of net $CO_2$ uptake are very similar at the two sites (Fig. 5b). Still, over time, deviations in uptake rates become obvious. For instance, by the end of the 2015 sub-season the drained site showed
a net uptake of ~ 20 $gCO_2$-C $m^{-2}$ less than the reference site. The analysis of component fluxes (Fig. 6b) indicates that the difference in NEE can be attributed to the higher release of $CO_2$ via ecosystem respiration at the drained site, while photosynthetic uptake is close to uniform between sites for all years (Fig. 6b).

3.  Late season: Early-season differences in NEE between the drained and reference sites continue during the late-season. With the onset of vegetation senescence both areas shift from a net sink to a net source of $CO_2$ (Fig. 5c), but
for all data years the reference site has a higher cumulative uptake of $CO_2$ from the atmosphere. The timing of the peak cumulative budget (i.e., the date when the ecosystem shifts from a net sink to a net source of $CO_2$), differs by year, and seems to be strongly influenced by air temperature conditions in August, with the latest peaks coinciding with the warmest temperatures (data not shown). Differences in cumulative NEE between the two treatments amount to ~ 11 (± 4) $gCO_2$-C $m^{-2}$; again, the effect was primarily driven by higher respiration rates in the drained
area. The observed slight difference in GPP between the drained and reference sites was non-uniform between years (Fig. 6c).

4.  Post-season: Differences in the cumulative carbon budgets between both sites further increased, as evidenced by the consistently higher cumulative values presented for the drained site (Fig. 5d). Both sites are continuous sources for



$CO_2$ during this sub-season, with decreased incoming radiation and air temperature conditions leading to the senescence of the last remaining active vegetation patches. This results in near-zero assimilation fluxes, and thus no diurnal NEE cycle. Higher respiration rates in the drained area lead to higher net $CO_2$ emissions (Fig. 6d).

Across all of the sub-seasons, our results demonstrate that the reference site was a stronger sink for $CO_2$ than the drained site. The majority of the observed $CO_2$ flux differences are caused by higher ecosystem respiration rates at the drained site, especially during the early- and late-seasons, while only negligible differences in GPP are observed. Over the entire growing season, the cumulative uptake at the reference site is 20–50 $gCO_2$-C m$^{-2}$ higher than at the drained site, with significant differences (ANOVA, $p < 0.05$) between both treatments in both 2013 and 2015 (in 2014 no significant difference in cumulative NEE between the sites was found). Daily mean flux rates over the course of the 2015 growing season are shown in Fig. A1a.

### 3.2 Comparison of the recent (2013–2015) and historic (2002–2005) datasets

Data coverage between the historic and recent observation periods varied considerably, with measurement-starting dates ranging from the beginning of May to mid-July, and with end dates differing by up to 2 months (Fig. 2). To facilitate a direct comparison of the carbon budgets between all years, the starting dates for cumulative time series (Fig. 7 & Fig. 8) were set to first of May. For time series that started later in the season, we used the available budget from the nearest year (either prior to or after the year of observation). Accordingly, not all displayed time series start with a value of zero, and final values therefore do not always represent the cumulative budget based on the available observations. This approach facilitates a qualitative comparison of seasonal trends in $CO_2$ exchange between different data years, while absolute numbers of the seasonal budgets are subject to significant uncertainty for years with data gaps.

The observed differences in cumulative uptake for the historic dataset reflect the interannual variability in $CO_2$ fluxes that can be expected in this type of ecosystem (Fig. 7a, 2002–2004). Such changes are primarily caused by varying climate conditions from year to year (Merbold et al., 2009). In contrast, 2005 - the year immediately following the installation of the drainage system - shows a systematic dampening in the seasonal course of net carbon exchange (Fig. 7b), even though the cumulative NEE budget at the end of the growing season is comparable to previous years' levels.

A direct comparison of the recent and historic carbon flux patterns in Fig. 7 shows that the seasonal amplitude of the cumulative $CO_2$ budget has increased over the past decade. Maximum $CO_2$ uptake rates are higher at peak growing season in July, and initial carbon release immediately following snowmelt (~ May–June) seems to have increased as well. In contrast, net $CO_2$ emission rates in autumn appear to be similar between recent and historic datasets. As a consequence, the average cumulative growing season uptake of $CO_2$ by the ecosystem has grown over the past ~ 10 years.

While the range of the cumulative $CO_2$ budget has partial overlap between the recent and historic data years for reference conditions, the fluxes observed immediately after the disturbance (2005) and recent fluxes from the drained area display systematic differences, both in absolute values as well as in seasonal patterns. This finding demonstrates that the disturbed area has strongly rebounded from the immediate drainage effects and the resulting dampened seasonality that



occurred shortly after the disturbance. Instead, recent $CO_2$ uptake rates from the drained section of the tundra partly exceed the rates observed for the reference area before installation of the drainage (Fig. 7, 2002–2004).

The seasonal $CO_2$ exchange for the overlapping data period (19 July  to 23 September) for the years 2002–2004 summed up to  -43, -22, and -17 $gCO_2$ C m$^{-2}$, respectively, and -41, -63 and -42 $gCO_2$ C m$^{-2}$, respectively, for the years 2013–2015 under reference conditions. For the drained part of the tundra, we found -6 $gCO_2$-C m$^{-2}$ for 2005 and -19, -47 and -11 $gCO_2$-C m$^{-2}$ for 2013–2015, respectively. Negative budgets over most of the growing season in all observation years indicate that both the drained and reference ecosystems act as net sinks for atmospheric $CO_2$ during this part of the year. However, the drained ecosystem displays slightly different exchange patterns for $CO_2$, with higher respiration rates

throughout the growing season, making it a weaker $CO_2$ sink in comparison to the reference site.

Separation of NEE into its individual component fluxes - GPP and Reco - demonstrates that the interannual variability in both cumulative time series (recent: 2013–2015 and historic: 2002–2005) increased strongly after mid-July. Net flux variability is dominated by changes in photosynthetic uptake (Fig. 8b). Absolute cumulative flux rates of GPP and Reco tend to be higher at the drained site, indicating that the carbon turnover rates have increased as a consequence of drainage. The

long-term tendencies toward higher net $CO_2$ uptake of the ecosystem during the summer are mainly driven by increases in GPP since overall increases in photosynthetic uptake outweigh higher respiration losses.

### 3.3 Linking interannual shifts in carbon fluxes to phenology

We found a strong interannual variability in timing for key phenological stages in the Chersky area (for the MODIS period, 2000 to present) with respect to the start (SOS), peak (POS) and end of the season (EOS, Fig. 3). Maximum year-to-

year variability ranged between 18, 26 and 6 days for SOS, POS, and EOS, respectively. Focusing exclusively on the observation periods 2002–2005 and 2013–2015, we found a maximum difference of 18 days between the shortest (118 days, in 2015) and longest (136 days, in 2005) growing season lengths. Neither the timing nor duration of seasons showed any long-term trends, resulting in no detected systematic differences in phenology between the historic and recent observation periods.

Our limited data availability precludes establishing significant links between phenology and NEE data. The analysis of the interannual variability of cumulative $CO_2$ flux budgets was executed only for the reference site, limiting the data pool to just six years (2002–2004 and 2013–2015). Moreover, observations made during the historic period did not cover the entire growing season in some years, further reducing the database for the purpose of statistical analyses. It was therefore not possible to identify meaningful links between individual key dates and NEE patterns. Similarly, no direct correlation

between growing season length and NEE budgets in either the recent or historic datasets was found.

Normalizing $CO_2$ time series to account for shifts in phenology (see Fig. 3 and Sect. 2.6) removes interannual variability in cumulative flux budgets and separates the cumulative $CO_2$ budgets into two distinct groups, the recent (2013–2015) and the historic (2002–2004) datasets (Fig. 9). Differences in $CO_2$ flux budgets become particularly obvious in the early- and late-seasons (Fig. 9 b and c). A linear regression fit to identify the maximum gradient of the growing season $CO_2$





uptake (Table 2) confirms this separation, indicating a significant increase in this peak NEE uptake (ANOVA, $p = 0.007$)

between historic and recent datasets. However, the timing of the maximum uptake (as DOY of maximum gradient, Table 2)

seems not to be influenced, as dates vary from mid to late July without any clear pattern. For this gradient analysis we tested

different sizes of the moving window, and found no sensitivity of the obtained results towards this parameter. During both

the pre- and post-season the net $CO_2$ release signal shows only minor differences between the recent and historic data (Fig. 9

a and d).

## 4 Discussion

### 4.1 Impact of decadal drainage

In Arctic tundra ecosystems reduced soil moisture and the related decrease in water table depth increase the

decomposition of soil carbon. This stems from the increased diffusion of oxygen into the soil (Billings et al., 1982; Oechel et

al., 1998; Peterson et al., 1984), which induces higher $CO_2$ exchange fluxes to the atmosphere. On the other hand,

partitioning of the NEE signal indicates a minor increase in GPP rates at the drained site, which can be attributed to a shift in

the vegetation composition. Drainage has led to the establishment of more tussock-forming sedges (*Carex* species) and

shrubs (*Betula exilis* and *Salix* species), while the undisturbed parts of this tundra ecosystem are instead dominated by cotton

grasses (*Eriophorum angustifolium*, Kwon et al., 2016). Such shifts in vegetation community structure have been associated

with differing $CO_2$ uptake patterns (Christensen et al., 2000). Major differences - which were especially clear in the early-

and late-season, when emissions from the drained site exceeded those found at the reference site - are caused by higher

respiration activity, triggered by higher soil temperatures (Fig. A1c) in the upper part of the active layer (Lloyd and Taylor,

1994).

In an earlier study, Merbold et al. (2009) could not show a significant change in the $CO_2$ exchange patterns immediately

following this drainage disturbance. As this result can be attributed to the shorter observation period of only a single data

year, neither long-term effects nor interannual variability could be taken into account. In contrast, in the present study we

demonstrate that long-term (>10 years) drainage had a significant impact on the carbon cycle processes of this tundra

ecosystem. Since recent $CO_2$ uptake rates from the drained site at times exceed the rates observed for the reference area

before installation of the drainage, we conclude that the drained section has largely adapted to the new conditions, with

features such as vegetation and microbial community structures that can tolerate lower average water table depths and

stronger fluctuations in water availability over the course of the growing season (Kwon et al., 2016). Concerning interannual

variability in the fluxes, decomposition of soil organic matter seems to be relatively unaffected by annual shifts in either

climate or water conditions, while assimilation can vary substantially from year to year. This variability in GPP is primarily

due to changing temperatures and cloud cover, and is further amplified under the influence of fluctuating water regimes at

the drained site. Finally, despite the fact that long-term changes in site conditions (e.g., water availability, plant



communities) can lead to systematic shifts in both absolute fluxes and flux components, we found interannual variability to remain at approximately the same level between historic and recent data.

To our knowledge, our study represents the only long-term experiment focusing on the net impact of lowered water table depths on the carbon cycle in Arctic tundra ecosystems. Existing water manipulation approaches have been conducted
that can be used to evaluate our findings, despite the fact that differences between wet and dry sites , respectively, were obtained by wetting the natural tundra, instead of drying as in case of the presented study. For example, in one study, long-term water table depth increase triggered a stronger $CO_2$ sink during the growing season by promoting photosynthetic uptake during the summer (Lupascu et al., 2014). Other studies found lower $CO_2$ emissions during autumn as a consequence of wetter conditions (Christiansen et al., 2012). Similar conclusions were drawn in the present study, where wetter conditions
under undisturbed conditions led to lower respiration rates and a higher $CO_2$ sink compared to the drained site. However, a direct comparison of our results with prior studies is compromised due to the different measurement techniques used, as well as the multi-factorial interaction of different treatments combined with the increased water table depth.

Similar findings to these long-term experiments have been described by studies that focus on short-term processes using an eddy-covariance method. Dry conditions are considered to be an important control factor for the terrestrial carbon cycle,
and have been linked to increased respiration rates during the growing seasons (Aurela et al., 2007; Huemmrich et al., 2010; Lund et al., 2012; Zona et al., 2010). Such findings are in line with the results we obtained shortly after the setup of the drainage channel (2005). Although Zimov et al. (1996) observed the highest effluxes during the wettest year in an area close to our site, these results may have been influenced by small scale heterogeneity and differences in measurement technique (i.e., chamber vs. eddy-covariance).

**4.2 Seasonal development and interannual variability of $CO_2$ fluxes**

The methods and limits used in determining the key dates for seasonality have a high degree of uncertainty. This uncertainty can influence the key dates derived to indicate SOS, POS, and EOS, which can in turn lead to uncertainty in the annual and sub-seasonal NEE budgets. Variable spring flooding can also affect the duration of the growing season through an indirect shift in key dates, and can reduce $CO_2$ uptake during the growing season (Zona et al., 2012). However, as we do
not expect potential uncertainties in the key dates to cause systematic shifts in seasonal trends and budgets, the qualitative results presented in the present study should not be affected.

Pronounced net $CO_2$ emissions to the atmosphere at the start of the growing season, as observed in this study during the pre-season, have previously been described by other studies for Arctic tundra sites (Corradi et al., 2005; Euskirchen et al., 2012; Grøndahl et al., 2007; Kutzbach et al., 2007; Nordstroem et al., 2001; Oechel et al., 2014; Oechel et al., 1997;
Parmentier et al., 2011; van der Molen et al., 2007; Zimov et al., 1996). During the pre-season, respiration clearly dominates the $CO_2$ flux signal, while the onset of photosynthetic uptake is delayed even though environmental conditions are adequate (e.g., mean temperature = 8.1 °C, mean shortwave incoming radiation = 170 W m$^{-2}$). This observation, which differs




markedly from the results of other studies focusing on Arctic tundra $CO_2$ flux exchange (Harazono et al., 2003; Kutzbach et al., 2007), might be associated with the very low abundance of mosses in this specific ecosystem (Kwon et al., 2016).

The interannual variability of cumulative $CO_2$ fluxes is weak during the pre-season, even if environmental conditions (i.e., air and soil temperatures, soil moisture) are highly diverse both within and before the individual pre-seasons. For example, 2004 showed exceptionally high air temperatures during this sub-season, while net $CO_2$ emissions remained at the same levels as had been observed for years with more regular temperature courses (Fig. B2). In contrast, in 2015 release rates were higher than average during a relatively cold pre-season. We were unable to correlate the early start of the growing

season and warm springtime conditions with increased uptake patterns during the growing season as described by other studies such as Aurela et al. (2004) and Euskirchen et al. (2012). These findings suggest that, for our study site near Chersky, early-season emissions may be affected by both conditions during winter (i.e., snow cover and depth, onset of snowmelt) and transition period (i.e., start and intensity of the flooding event). A detailed effect cannot be quantified here since all relevant snow regime dynamics happen before the start of the pre-season. Based on the available data, no direct correlation between

the years with the lowest uptake over the growing season and the strongest flooding event were observed; an isolated effect of the flooding event therefore cannot explain the observed interannual variability.

Both peak uptake rates and the timing of these peaks at the end of the early-season (i.e., mid to late July) agree with findings presented for tundra systems across the Arctic circle (Euskirchen et al., 2012; Grøndahl et al., 2007; Harazono et al., 2003; Kutzbach et al., 2007; Lafleur and Humphreys, 2007; Nordstroem et al., 2001; Oechel et al., 2014). During the late-

season the shift of the balance between respiration and assimilation towards a net release of $CO_2$ is highly variable between years. Results from 2014 indicate that the length of the net uptake season is highly sensitive towards climate conditions in July to August. In 2014, warm temperatures (> 5 °C compared to the average monthly mean in August) led to an increased net $CO_2$ uptake from mid-July to the end of October by ~ 20 gCO$_2$-C m$^{-2}$.

The year-to year variability in NEE during the early- and late-seasons is dominated by variability in GPP (Euskirchen et

al., 2012; van der Molen et al., 2007). In our study, we found GPP also to be the main driver of shifts in summertime $CO_2$ fluxes between the recent and historic datasets. The observed net increase in the $CO_2$ sink strength in recent years can be attributed to significant increases in $CO_2$ uptake rates around the peak of the growing season (Table 2). This observed separation in the cumulative $CO_2$ patterns (Fig. 9) between historic and recent periods cannot be attributed to systematic changes in climate: There are no systematic trends in air temperatures over the last decade in the Chersky region, and,

particularly during the early- and late-season, average conditions were found to be very stable over the years (Fig. B1). Small deviations in temperature during pre- and post-seasons are not correlated with the observed trends in $CO_2$ uptake. On the other hand, we observed a minor increase in cumulative NDVI values (Fig. B2) in recent years (2013–2015), and these higher photosynthetic leaf areas might indicate an overall increase in light use efficiency within this ecosystem. Since the MODIS NDVI data result from a mixed product covering both the reference and drained sites, values might have changed

slightly due to changes in the vegetation composition at the drained site (Kwon et al., 2016). However, considering these





small differences without a trend between recent and historic years, this data product alone cannot provide a final answer regarding overall shifts in vegetation structure.

Another possible factor influencing the differences observed between recent and historic datasets is the location of the reference site. Since the reference site from the historic dataset was converted into a drained site in Autumn 2004 with the installation of the drainage ditch, a new reference site had to be chosen for the recent measurements. Elevation measurements in 2014 indicated that parts of the drained site are slightly lower than the surrounding terrain (up to ~ 0.5 m). Within the floodplain environment, such minor topographic depressions preferentially collect melt and flood water; accordingly, reference observations made during the historic period may have been subject to more pronounced flooding as compared to the recent reference observations at the new tower position. More standing water surrounding the tower site could in turn cause differences in photosynthetic activity, respiration rates, and NDVI. Flooding dynamics can also influence carbon exchange fluxes and increase $CO_2$ losses (Zona et al., 2012). However, differences in elevation might have been less pronounced in the historic dataset, since subsidence may have been a secondary disturbance effect triggered by the lowered water table depth in this area.

Similar year-to-year patterns of pronounced $CO_2$ efflux rates can be observed in the post-season, where temperatures remain above freezing in most soil layers with favourable conditions for the microbial decomposition, triggering enhanced $CO_2$ emissions (Parmentier et al., 2011). However, unlike to Oechel et al. (2014), respiration rates at our site are lower during post-season (~ 35 %) than during late-season. This is primarily due to biogeophysical differences in the site conditions in Chersky as compared to the Alaska North Slope sites near Barrow. While thawing of the active layer continues in autumn near Atqasuk (Alaska, US), soil temperatures are already starting to decrease by August to September (see Fig. A1c and Table A1) in Chersky.

### 4.3 Data processing and uncertainties

The Burba correction was applied to adjust for the surface heating effects on the measurement performance of the open-path gas analyzer (LI-7500, LI-COR Biosciences); however, our approach is different from the original formulation of this procedure as detailed by Burba et al. (2008). Our approach was to customize the original equation for a non-vertical orientation of the instrument. Since our analyzer was mounted in an inclined position, the majority of the self-induced heating did not affect the measurement path; as a result, only a small fraction of the original Burba correction term needed to be applied (Jarvi et al., 2009; Rogiers et al., 2008). We used a dataset of closed-path fluxes as a reference (data not shown), and found that 7 % of the original Burba correction optimized the agreement between open-path fluxes and the reference closed-path fluxes. As this value is dependent on changes in the reference closed-path dataset (e.g., data processing or software), the fraction of the applied correction might still be subject to change in future experiments. Still, with data processing based on standardized methods (Fratini and Mauder, 2014), the estimated fractionation factor is comparable to that presented by Jarvi et al. (2009). A good correlation between open- and closed-path data after the correction ($R^2 = 0.92$;



slope = 1.05 mmol m$^{-2}$ s$^{-1}$ and offset = -0.0001 mmol m$^{-2}$ s$^{-1}$) was also obtained. We therefore assume that our correction is representative to remove systematic effects of sensor heating from the open-path flux data used in the present study.

The recent data coverage ranged between 60–80 % over the different years of eddy-covariance flux measurement using closed-path observations. Data coverage of the open-path time series was lower, with extra gaps due to unfavorable meteorological conditions (e.g., rain, fog, dew) that limited the ability of the chosen measurement technique (Haslwanter et al., 2009; Mauder et al., 2008). For this remote study site, failures in the power supply system can also cause frequent gaps, which was especially the case for the historic dataset, as there was no continuous or stable power supply during this time

(Merbold et al., 2009). For the recent dataset, gaps were commonly less than a day and distributed equally over all sub-seasons. Longer gaps during the recent observation period were caused by flooding events, when the entire system needed to be shut down to avoid damage, and for a single event as a result of an additional laser offset in the closed-path gas analyzer at Tower 1 in 2015. Since the longest gaps have been present during pre-season - a critical transition period between winter and the growing season when environmental conditions change rapidly - even relatively short data gaps can have an impact

on the final time series of $CO_2$ fluxes. For instance, as seen in the 2015 Tower 1 dataset, a poor performance of the gap-filling algorithm produced a sharp transition between winter (with near-zero fluxes) and the pre-season (with pronounced release, Fig. 8b). However, for long-term budgets, this short period with relatively low flux rates is not relevant, and during the remaining sub-season only short gaps occur, during which the algorithm performs well.

The flux-partitioning algorithm that splits NEE into GPP and Reco signals is based on the estimation of the nighttime

respiration as a function of temperature and the extrapolation of this relationship for the daytime (Reichstein et al., 2005). In a subsequent processing step, GPP is derived from the difference between NEE and Reco (for details see Reichstein et al., 2005). The results for the component fluxes GPP and Reco presented here are therefore not based on direct measurements, but rather represent a model-data hybrid product. The performance of this procedure is dependent on a number of user-defined settings, such as the definition of nighttime conditions by setting a radiation threshold (present study: 20 Wm$^{-2}$;

previously, e.g., Parmentier et al., 2011; Runkle et al., 2013). To address Reco's temperature dependence, we used air temperature instead of soil temperature. This decision was based on our observance of implausible patterns in near-surface temperature measurements at our site, and as, because of a damped diurnal cycle, higher-quality soil temperatures at 0.08 m below the surface cannot be correlated to short-term changes in respiration patterns. During the growing season, air and near-surface soil temperatures are closely linked, and both represent the temperature regime within the soil, driving respiration

processes (Lloyd and Taylor, 1994). However, this flux-partitioning approach using air temperature data is limited during the thawing and refreezing periods when energy is not used for heating, but for the phase change of ice-water, and when soil temperatures do not change relative to air temperatures. To account for this effect, we executed flux-partitioning algorithms separately for each sub-season. With this approach we ensured that our method is based on adequate relationships between temperature and respiration rates at each point in the growing season. This seasonal separation also prevents potential

overestimation of base respiration (Mahecha et al., 2010), which is related to growth in both microbial and plant biomass with increasing thaw depth and rising soil temperatures. Overall, the output of such a flux-partitioning algorithm is subject to



increased uncertainties based on the chosen methodology and user-defined settings; the results on the component fluxes presented in the present study must be interpreted accordingly.

The comparability of the flux datasets between observation years has some technical limitations due to changes in instrumental setup (eddy-covariance equipment). For the historic dataset, both open-path (LI7500) and closed-path (LI6262) infrared gas analyzers (IRGAs) were used, while recent data are based on an open-path (LI7500) IRGA and a different type of closed-path analyzer (Los Gatos FGGA, off-axis integrated cavity output spectroscopy). Differences in the analyzer type also require different data post-processing and correction, and can result in deviations in the final flux rates even if the observed signals are similar. When integrating eddy-covariance fluxes over longer time periods (e.g., weeks, months, years), previous studies indicate that averaged cumulative NEE tends to be more positive when results are based on $CO_2$ signals from an open-path system (Bowling et al., 2010; Clement et al., 2009; Haslwanter et al., 2009). Even though most of these studies have been based on a newer version of the closed-path instrument (LI7000), their results suggest that part of the differences found between the historic and recent datasets presented in the present study might be related to changes in the instrumental setup.

## 5 Conclusion

Our results demonstrate that a long-term drainage experiment (> 10 years) in a permafrost floodplain in northeastern Siberia has significant effects on growing season $CO_2$ exchange patterns in comparison to a reference site reflecting undisturbed conditions:

1. Drainage effect on recent $CO_2$ fluxes: Our observations show that the net $CO_2$ sink strength of the drained ecosystem was reduced by 20–50 $gCO_2$-C $m^{-2}$ per year over the three observed growing seasons (2013–2015). Differences in NEE between the drained and reference ecosystems are dominated by increased respiration rates at the drained site, primarily triggered by higher near-surface temperatures. In contrast, we found only minor effects on photosynthetic uptake, and the differences between the drained and reference sites were less systematic across data years. Still, both ecosystems show negative flux budgets over the growing seasons, representing a net sink of $CO_2$ from the atmosphere.

2. Longer-term trends in $CO_2$ budgets: The comparison of recent flux data to historic conditions (2002–2005) indicates that this tundra ecosystem is now a stronger sink for $CO_2$ over the summer months than it was about 10 years ago. This finding can be linked to an increase in seasonality, with higher net emissions after the snowmelt, as well as to strongly intensified uptake rates at the peak of the growing season in July. The observed long-term shifts are dominated by higher photosynthetic uptake and a resulting stronger carbon sink capacity for $CO_2$ over the whole growing season, most likely linked to a trend towards higher plant biomass.

3. Ecosystem adaptation following disturbance: Our results demonstrate that the drained ecosystem did recover from the intermediate disturbance impact in 2005 (directly after the installation of the drainage), when net carbon




exchange rates showed a systematically damped seasonal course. $CO_2$ flux data from recent years emphasize that adaptations to the changes in biotic (e.g., vegetation community) and abiotic (e.g., soil thermal and water regimes) site conditions have resulted in a rebounding of the carbon cycle processes to previous flux levels of the ecosystem.

4. Seasonality in summertime $CO_2$ exchange processes: Concerning recent effects of the drainage disturbance, 70 % of the observed differences in cumulative $CO_2$ budgets can be attributed to conditions in the high-summer period (July–August), when the effect of reduced water availability is most pronounced. Concerning long-term trends, the major part of the differences between historic and recent flux budgets was found in the early-season, when the long-term increasing trend in photosynthetic uptake has the strongest effect.

**Acknowledgements and author contribution**

This work was supported by the European Commission (PAGE21 project, FP7-ENV-2011, Grant Agreement No. 282700, and PerCCOM project, FP7-PEOPLE-2012-CIG, Grant Agreement No. PCIG12-GA-201-333796), the German Ministry of Education and Research (CarboPerm-Project, BMBF Grant No. 03G0836G), the AXA Research Fund (PDOC_2012_W2 campaign, ARF fellowship M.Göckede), and the European Science Foundation (ESF for the activity "Tall Tower and Surface Research Network for Verification of Climate Relevant Emissions of Human Origin", Short Visit Grant, fellowship F. Kittler). The authors appreciate the efforts of NESS staff members, especially Galina Zimova and Nastya Zimova, for organizing field work; they also recognize the team from the Field Experiments & Instrumentation group (MPI-BGC), especially Martin Hertel, for supporting field work. We thank Thomas Foken and Werner Eugster for scientific input and discussion.

We applied first-last-author-emphasis and equal-contribution (alphabetical sequence) methods for the order of authors (Tscharntke et al., 2007).

**Appendix A: Recent environmental conditions (2013–2015)**

Large differences in all meteorological site conditions were found for all sub- seasons and over the entire year (Table A1). For example, in 2014 the mean annual temperature was -10 °C, with daily means ranging from -49 °C in January to +22 °C in July. Early- and late-season are the warmest periods of the year for both air and soil temperatures, with particularly high values observed during late-season 2014. Long-term air temperature averages (Rohde et al., 2013) for August 1982–2011 (data not shown here) indicate conditions above the monthly mean for 2014 and below the monthly mean for 2013 and 2015.

A pronounced seasonal cycle due to freezing-thawing dynamics is observed in the soil temperatures (Fig. A1c). For both sites, the active layer was completely frozen until early June, with thawing gradually beginning from the top down. During the growing season the active layer increases up to a depth of ~ 0.4–0.5 m (data not shown). Refreezing usually starts



at end of September, and proceeds both from the surface downwards and permafrost upwards. For both the frozen and
unfrozen conditions the drained site had higher soil temperatures (Fig. A1c). The largest differences during the growing
season occurred during July and August with a ~ 2.5 °C higher soil temperature at the drained site.

        While precipitation is variable throughout the year, over 80 % of the annual budget was collected during the
growing season in our datasets for 2014 and 2015. Rain events are strongest during early- and late-season (Table A1 and Fig.
A2b) with events ranging from 0.1 mm d$^{-1}$ to 16 mm d$^{-1}$. These events have a direct influence on soil moisture (Fig. A2b).
After flooding abates soil moisture drops rapidly, especially in the tussocks (data not shown). During rain events the soil
moisture can increase to maximum values, depending on the intensity of the rain event. Soil moisture values are lower at the
drained site than under natural conditions; this can be explained by the ~ 0.2 m difference in water table depth (over and
under the soil surface for reference and drained ecosystems, respectively) between treatments (Kwon et al., 2016). While
values for both 2013 and 2015 (Fig. A1b) indicated very dry conditions under disturbed conditions, saturation during late-
season 2014 (data not shown) at both towers had the same value (soil water content at 0.08 m depth).

        Net radiation is dominated by strong changes in the albedo and pronounced differences in short wave radiation.
Short wave radiation varies enormously, with 24h during polar summer, none during polar winter, and transition periods
between. Levels of net radiation during the overlapping period of 2013–2015 (see Table A1) are comparable. Pre-season
data indicate that the highest net radiation values trigger plant growth, and that the rapidly decreasing radiation that occurs
during late-season fuels an already early senescence of the vegetation.

**Appendix B: Long-term environmental conditions**

        Variability in cumulative air temperature was observed, especially during pre- and post-season, while early- and
late-season showed more uniform patterns. The most notable findings were the warm conditions during the 2004 pre-season,
which, combined with a short sub-season, indicates a relatively early and intense start to the growing season. The same
values in at the end of the 2003 pre-season are attributed to a remarkably long sub-season. During the early- and late-season,
air temperature developed very similarly in all observed years, with different data years marking the minimum and
maximum thresholds. Extreme warm conditions are again observed in the 2002 and 2003 post-seasons, with values twice as
high as for the remaining years in the dataset.

        A subtle trend was observed in cumulative NDVI values (Fig. B2), with recent years at the upper range of the
values observed for the historic period. At the same time, all data years show very similar patterns within each sub-season,
with differences in values mainly attributable to the individual length of specific sub-seasons.

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





**Table 1:** Summary of the observed environmental variables available for both towers, including instrumentation and measurement height.
Tussock measurements describe the conditions in the middle of the tussock (0.2 m to the top of the tussock and 0.2 m to the soil surface).
Heights above the soil surface are indicated as positive numbers, depths below the soil surface are given as negative numbers.

| Environmental variable | Instrumentation | Measurement height [m] |
|---|---|---|
| Air temperature | KPK 1/6-ME-H38, Mela | 2 and 5 |
| Relative humidity | KPK 1/6-ME-H38, Mela | 2 and 5 |
| Barometric pressure | Pressure Transmitter, 61302V, Young | 5 |
| Long/short up/downwelling radiation | CNR4, Kipp & Zonen | 5 |
| Up/downwelling photosynthetically active radiation | PQS1, Kipp & Zonen | 5 |
| Precipitation | Tipping bucket rain gauge, Thies | 1 |
| Soil moisture | ML-2x, DeltaT | -0.08, -0.16, and tussock |
| Soil temperature | Pt100, YUMO | -0.04, -0.08, -0.16, -0.32, -0.64, -1.28, and tussock |



**Table 2:** Results from the linear regression of cumulative $CO_2$ fluxes during the growing season with day of year (DOY) of maximum gradient and maximum gradient slope.

| Year | DOY of maximum uptake | Maximum gradient |
|------|-----------------------|------------------|
| 2002 | 204 | -0.050 |
| 2003 | 195 | -0.044 |
| 2004 | 200 | -0.040 |
| 2013 | 210 | -0.058 |
| 2014 | 200 | -0.060 |
| 2015 | 191 | -0.064 |








**Table A1:** Environmental conditions measured at Tower 1 (drained): air temperature (Ta) at a 2 m height, soil temperature (Ts) at a 0.8 m depth and net radiation (Rn). Values are given as mean (± standard deviation), with the exception of precipitation (PPT, sum per season). Measurements from 2013 started mid-July (late-season); no data are therefore available before this time. The growing season is divided into sub-seasons, [length of season] = total number of days.

| Year | Season | Ta [°C] | Ts [°C] | Rn [Wm$^{-2}$] | PPT [mm] |
|---|---|---|---|---|---|
| 2013 | Late-season [33] | 8.7 (5.4) | 3.8 (1.1) | 80 (120) | NA |
| | Post-season [31] | 1.8 (4.8) | 0.9 (0.7) | 29 (82) | NA |
| 2014 | Year | -10.1 (18.9) | -2.8 (6.4) | 32 (103) | 173 |
| | Growing season [129] | 9.5 (7.4) | 3.8 (2.7) | 89 (137) | 153 |
| | Pre-season [27] | 8.6 (6.1) | 1.9 (1.7) | 143 (166) | 28 |
| | Early-season [36] | 14 (6.4) | 6.1 (1.5) | 121 (142) | 41 |
| | Late-season [35] | 12.1 (6.3) | 5.3 (1.8) | 80 (125) | 74 |
| | Post-season [31] | 2.2 (3.9) | 0.8 (0.9) | 19 (75) | 10 |
| 2015 | Year | -11.3 (18.7) | -2.4 (5.0) | 28 (96) | 118 |
| | Growing season [114] | 10.1 (6.8) | 3.3 (2.2) | 101 (141) | 99 |
| | Pre-season [19] | 8.1 (5.8) | 0.3 (0.9) | 148 (170) | 13 |
| | Early-season [44] | 14.9 (6.1) | 5.2 (1.5) | 126 (148) | 48 |
| | Late-season [34] | 7.7 (4.7) | 3.6 (1.1) | 72 (113) | 32 |
| | Post-season [17] | 4.9 (5.2) | 1.4 (0.7) | 39 (90) | 6 |




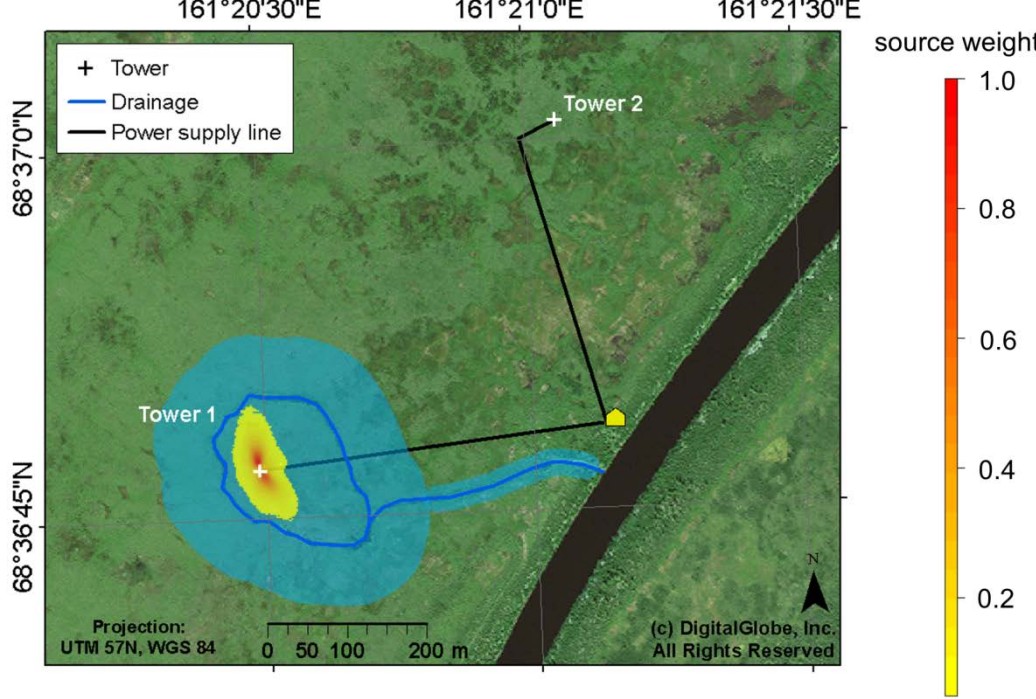

**Figure 1:** Overview of observational setup and site facilities on a wet tussock tundra near Chersky. The circular drainage ditch and drainage channel (blue lines) were installed in 2004 and are still in place today, altering the water table depth (blue shaded area outline the approximate area affected by the drainage). The image also depicts the towers (white crosses), power lines (black lines), and generator housing (yellow hut). The red-yellow colored area around Tower #1 indicates the weighted source function of footprints averaged over all stability classes from 15 May to 14 September 2014.




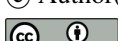

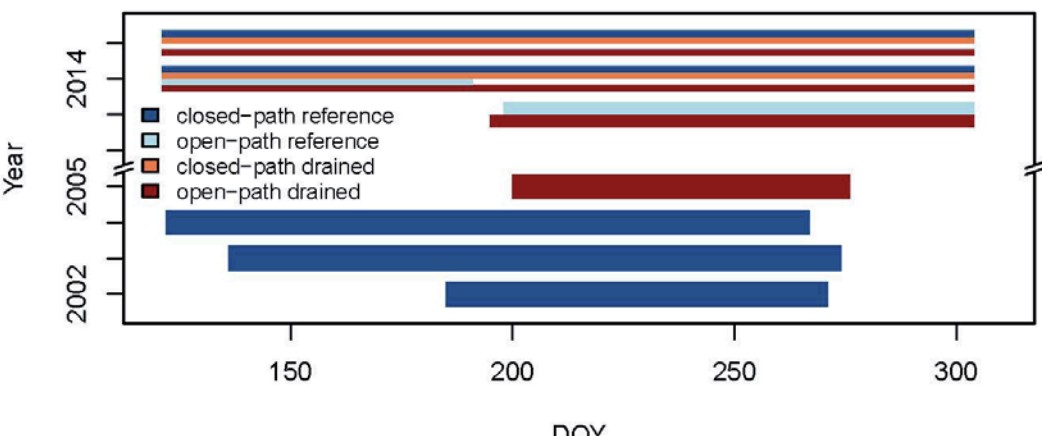

**Figure 2:** Measurement timeframe for the eddy-covariance systems (open-path = LI7500, closed-path = LI6262 and FGGA) of both towers, for both the recent (2013–2015) and historic (2002–2005) observation periods.





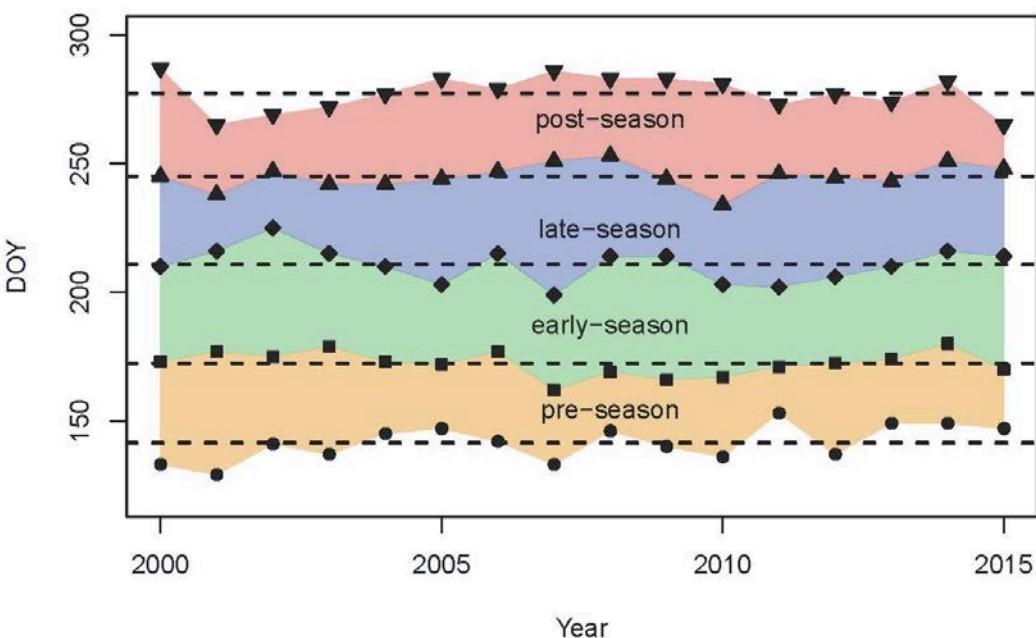

**Figure 3:** Interannual variability of all key dates with symbols marking the exact date, lines indicating the overall mean and colored areas representing the resulting sub-season during the growing season. Key dates from bottom top: start of pre-season (circle), SOS (rectangle), POS (diamond), EOS (standard triangle) and end of the post-season (upside-down triangle). Due to cloud coverage, in 2012 the MODIS data coverage was not high enough to determine the key dates (SOS–POS); these dates are therefore linearly interpolated.





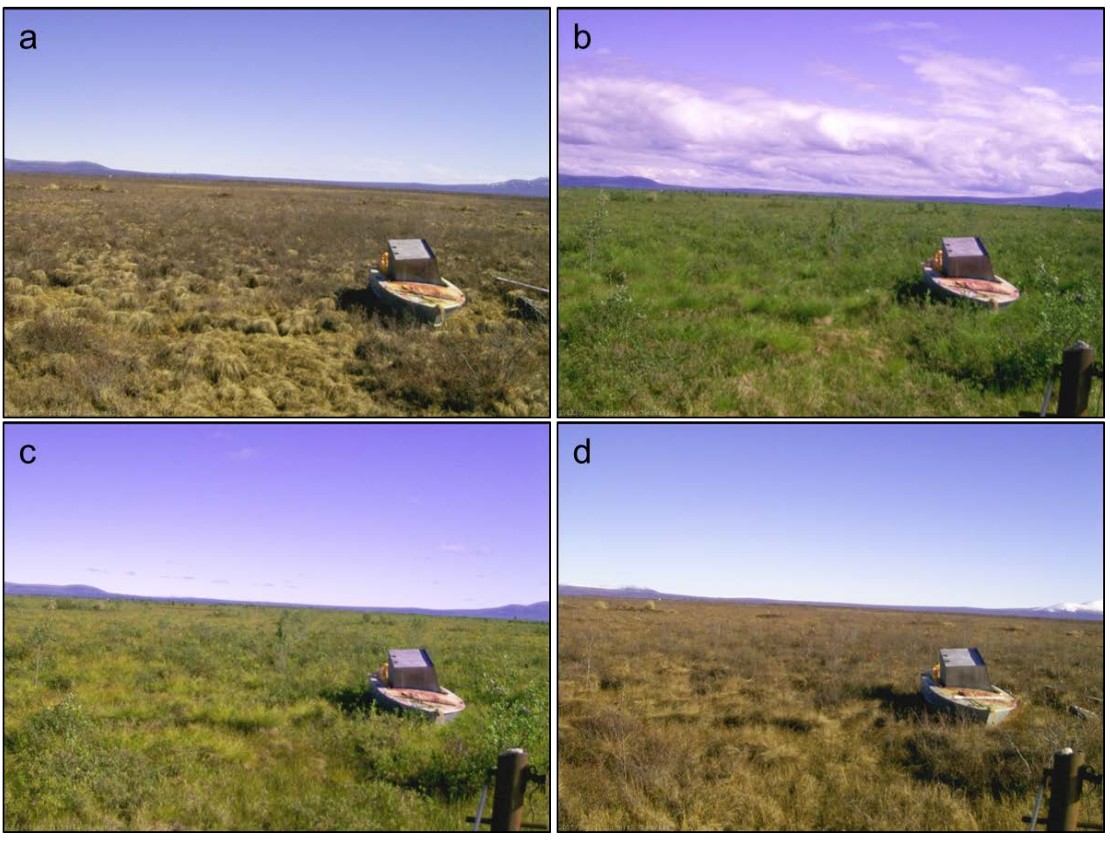

**Figure 4:** Webcam pictures representing the course of the 2015 growing season. All pictures were taken at noon in the middle of each sub-season: (a): mid pre-season (6 June), (b): mid early-season (10 July), (c): mid late-season (9 August) and (d): mid-post-season (21 September).





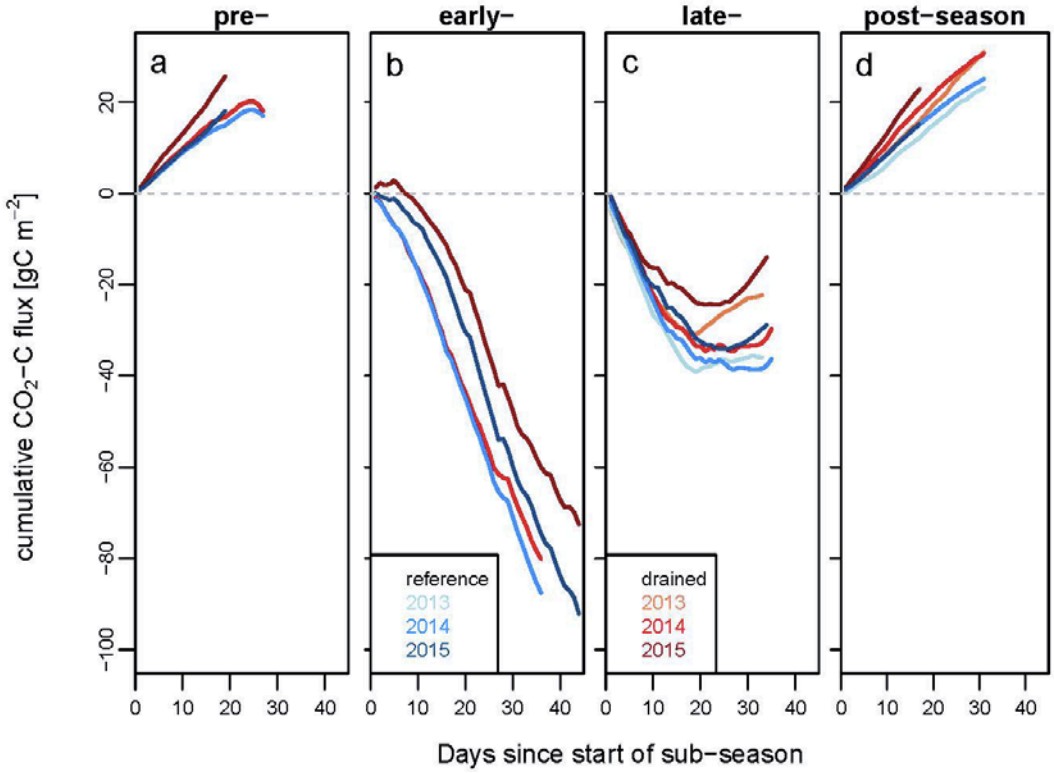

**Figure 5:** Cumulative $CO_2$ flux budgets for recent (2013–2015) eddy-covariance measurements for both sites (drained = red, reference =
blue). A grey line indicates the zero lines with a change from positive (i.e., release) and negative (i.e., uptake) fluxes. For these plots,
timestamps have been normalized to account for interannual shifts in phenology. Time series were separated into the four different sub
seasons (see Sect. 2.6) using individual key dates for each year. Time series were then plotted in separate panels, resetting the budgets to
zero at the beginning of each sub-season. No measurements are available for the 2013 pre- and early-seasons, since measurements started
mid-July 2013.





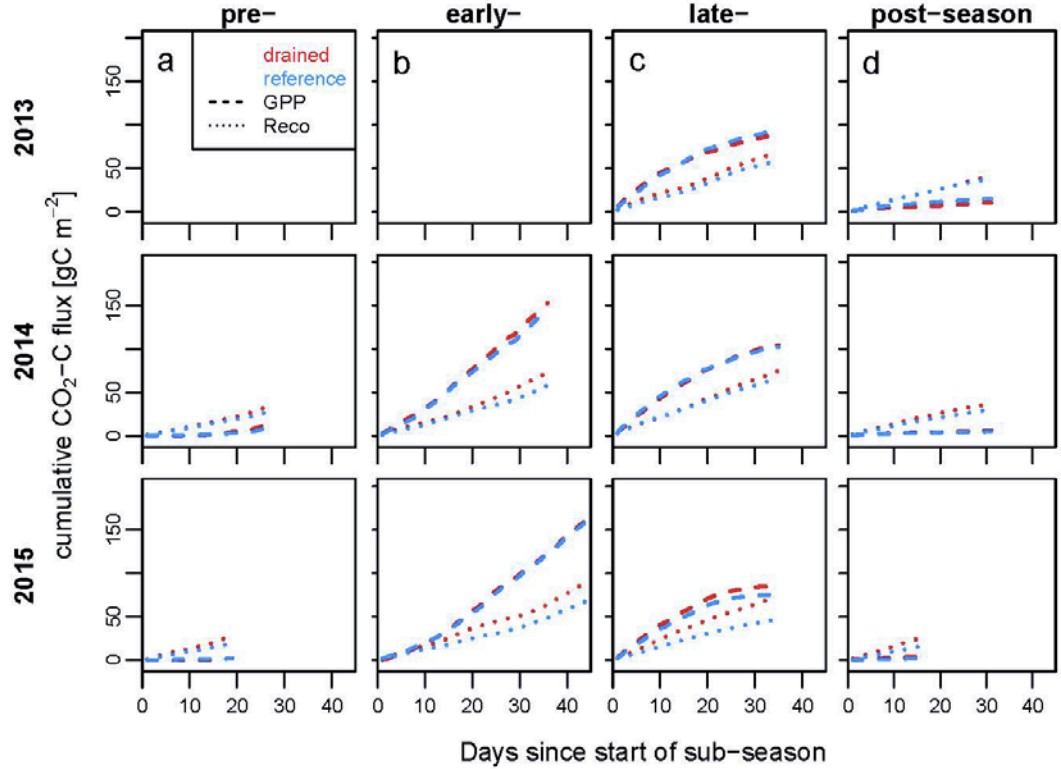

**Figure 6:** Cumulative ecosystem respiration (Reco, dotted line) and gross primary production (GPP, dashed line) from 2013 (upper panel) to 2015 (bottom panel) for both treatments (red and blue respectively as in Fig. 5). Time axes within this plotting scheme are identical to those used in Fig. 5.



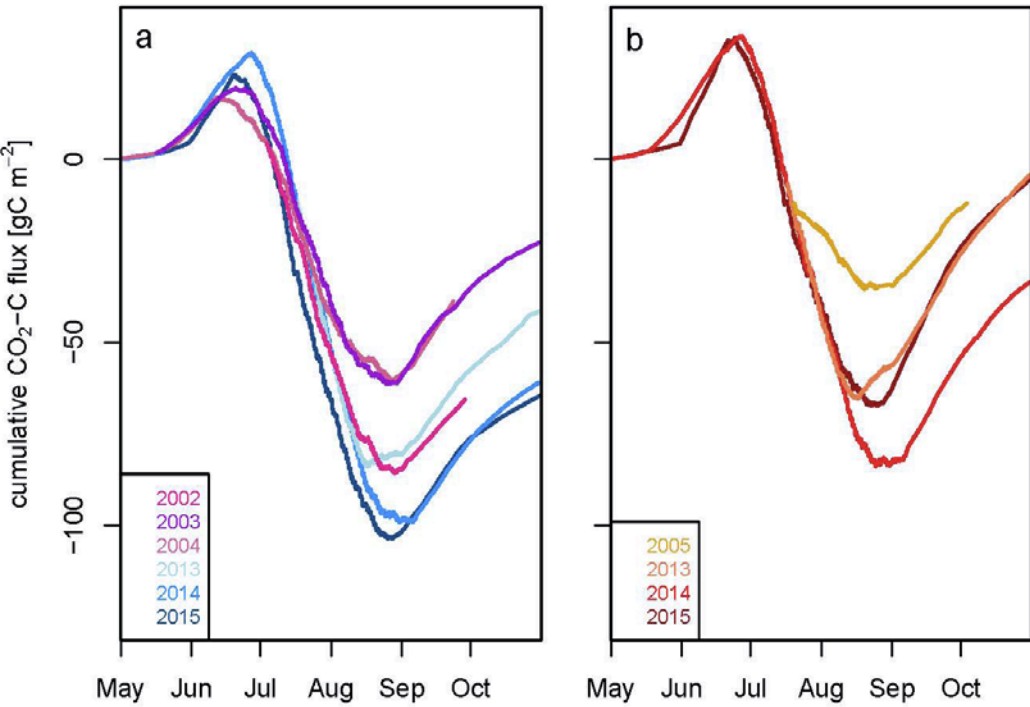

**Figure 7:** Recent (2013–2015) and historic (2002–2005) cumulative CO$_2$ budgets separated into disturbance regimes: (a): reference area; (b): drained area.



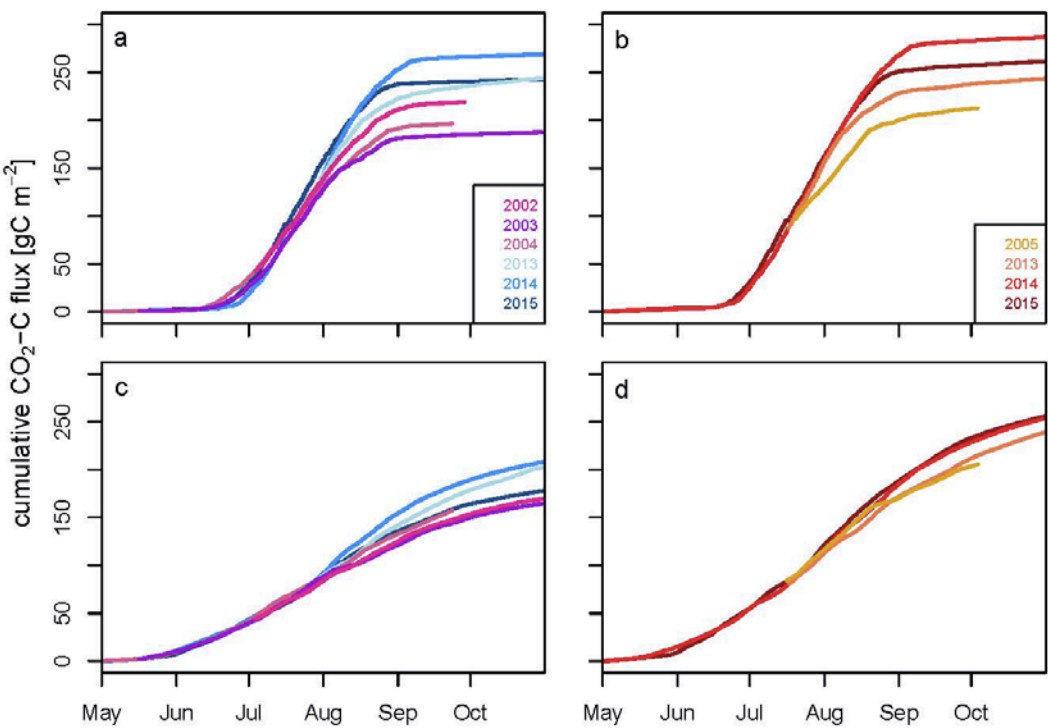

**Figure 8:** Cumulative signal of recent (2013–2015) and historic (2002–2005) gross primary production (GPP, top panel (a) and (b)) and ecosystem respiration (Reco, lower panel (c) and (d)), separated into disturbance regimes: (a) and (c): reference area; (b) and (d): drained area. Time axes within this plotting scheme are identical to those used in Fig. 7.





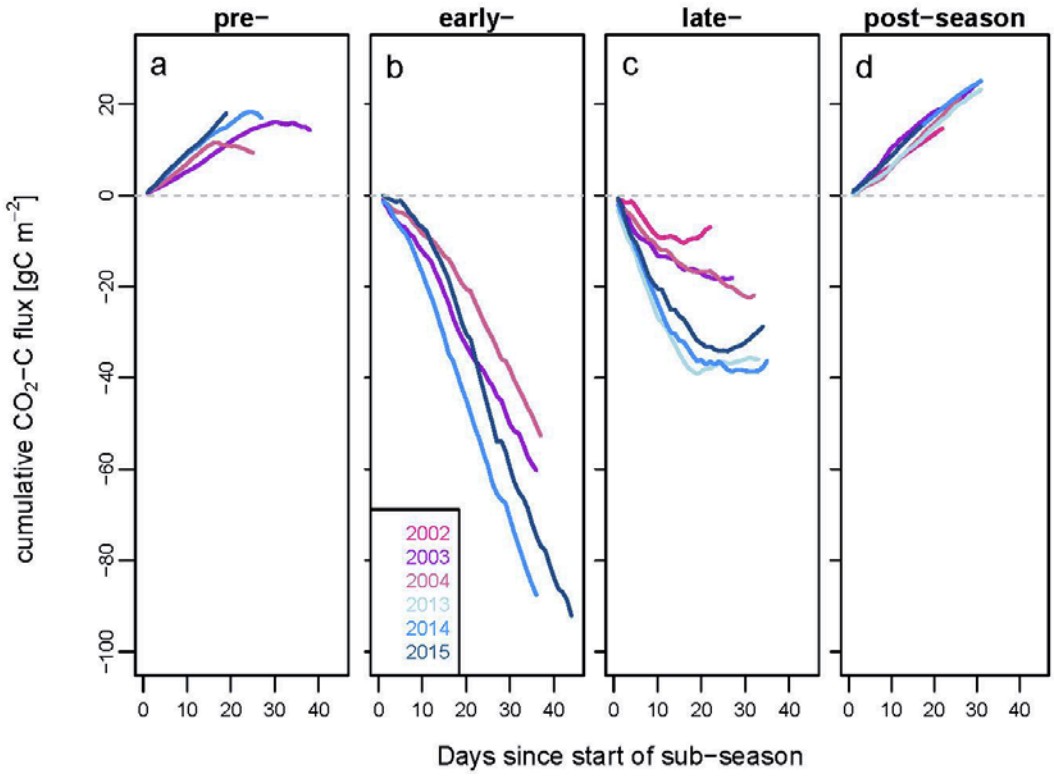

**Figure 9:** Sub-seasonal $CO_2$ flux budgets for the reference area for the historic (2002–2004) and recent (2013–2015) datasets. Time axes within this plotting scheme are identical to those used in Fig. 5.



**Figure A1:** Daily mean (a) carbon fluxes, (b) soil water content at 0.08 m depth and (c) soil temperature at 0.08 m depth. Since precipitation patterns are very similar between sites, only data from the drained site are displayed. Vertical lines divide the 2015 growing season into sub-seasons.




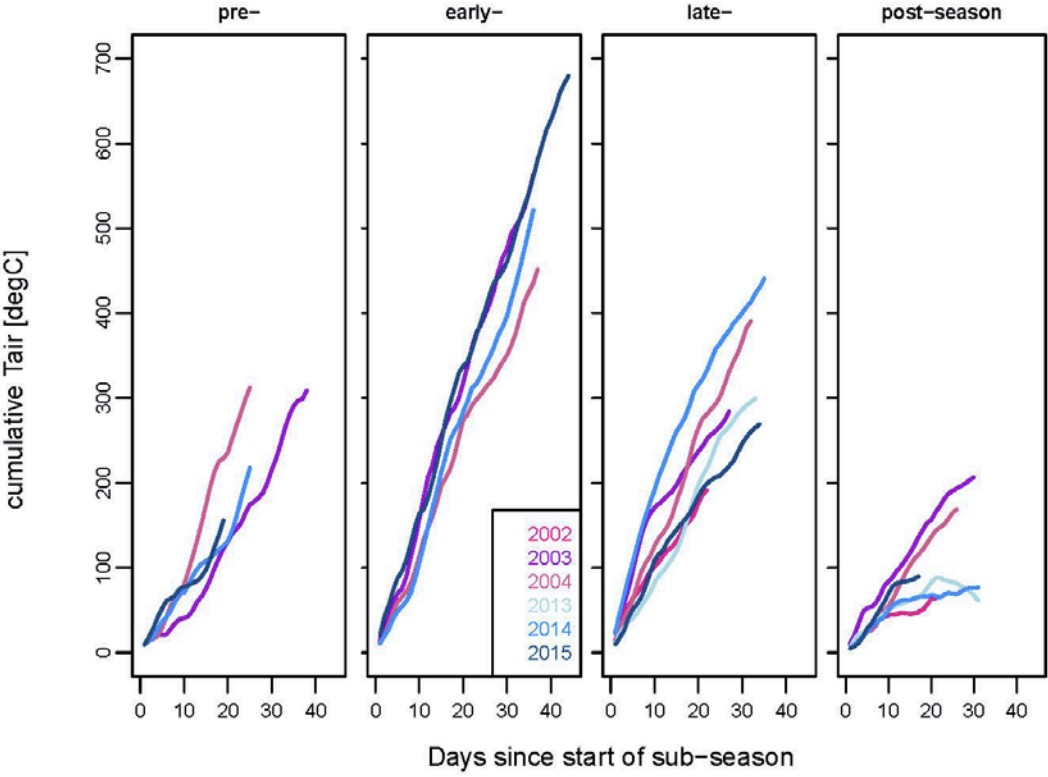

**Figure B1:** Sub-seasonal air temperature (at a 6m height) budgets for the reference area for the historic (2002–2004) and recent (2013–2015) datasets. Time axes within this plotting scheme are identical to those used in Fig. 5.




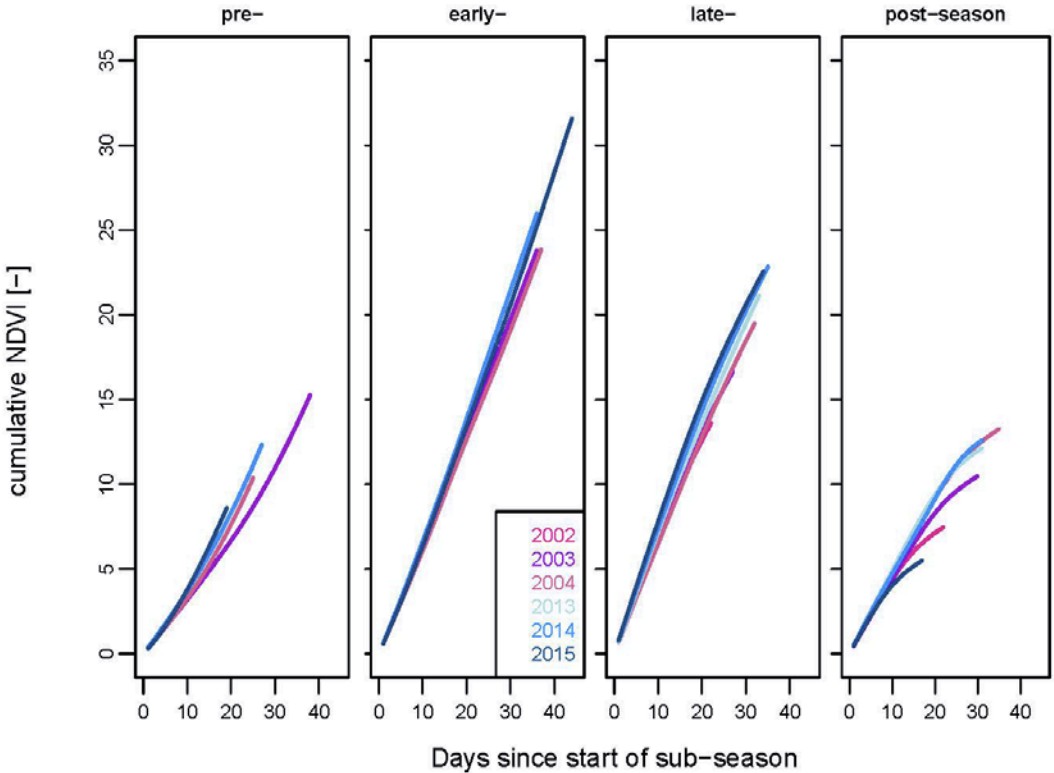

**Figure B2:** Sub-seasonal NDVI budgets for the reference area for the historic (2002–2004) and recent (2013–2015) datasets. Time axes within this plotting scheme are identical to those used in Fig. 5.