# Peer review of "Impacts of a decadal drainage disturbance on surface—atmosphere fluxes of carbon dioxide in a permafrost ecosystem"

_Biogeosciences, 2016_

## Referee Comment (RC1) · Anonymous Referee #1 · 25 May 2016

This manuscript examined effects of drainage on ecosystem CO2 exchange in tussock tundra ecosystem in Siberia. The research, which was conducted in an understudied and remote region of Siberia, has high relevance for understanding feedbacks from permafrost ecosystems to climate. The quality of the study and the analysis/presentation of results were strong, and the manuscript was well written. I have no major concerns about this publication, but I have suggested a number of minor edits and clarifications:

Line 16: clarify if you mean 'net CO2 uptake' or GPP

Line 22: it wasn't clear until I read the paper, what you meant by 'intensified'—may want to clarify

Line 30: I suggest updating to Hugelius (2014) and updated permafrost C numbers

Line 55: I don't think you mean methanotrophic here, which can occur under aerobic and anaerobic conditions. Change to methanogenic, if that's what you meant.

Line 60: There are quite a few long-term studies from Toolik—e.g., Mack et al (2004), Sistla et al (2013)

Line 116: I'm confused by 'during the first nine months' since the measurements only occur during the growing season.

Line 128: change 'data are collected' to data 'were'. Throughout methods, change to past tense.

Methods: Are missing a lot of information about gapfilling and partitioning the fluxes. But then I see this information is in the Discussion, which is a distraction from the discussion section and gap in the Methods. Most of that text in Section 4.3 should be moved to the Methods, the remaining could go in the Supplement.

Line 160: Change 'presented' to 'present'

Line 180: not sure what is meant by 'missing release'

Methods: Add in information about statistical design and analysis.

Line 207: I suggest to change 'evolving' to 'greening' or 'changing'

Line 215: change 'higher cumulative' to 'higher net cumulative'

Line 255-266: Given that there was only one historical year of data from the disturbed site, and given the level of inter-annual variability, I don't think the data support such a strong statement. I would change 'demonstrates' to 'suggests' and 'strongly rebounded' to 'may have rebounded'

Figures 7–9: Would be easier to see trends if you changed the line style for either the decadal or recent data so that one set is solid line and the other is dashed.

Line 190: ANOVA not mentioned in methods; as noted above, need to add in a statistical analysis section to the methods.

Lines 306-307: How was the effect of soil temperature, moisture, etc. analyzed?

Line 317: manuscript doesn't present organic matter decomposition results. Change to 'decomposition of soil organic matter' to 'ecosystem respiration'

Lines 323-33: The APEX peatland drainage experiment (Alaska) has been running since ~2005. Some of these pubs aren't long-term but they are water drainage studies in the permafrost regions, and their results may inform your discussion: Turetsky (2008, APEX), McConnell et al (2013, APEX), Natali et al. (2015, not APEX).

Lie 343: Change 'annual' to 'growing season'

Lines 378-379: Any long-term changes in snow depth? Or winter soil temps?

Line 485: What do you mean exactly by adaptations?

---

## Referee Comment (RC2) · Anonymous Referee #2 · 1 Jul 2016

Review of "Impacts of a decadal drainage disturbance on surface–atmosphere fluxes of carbon dioxide in a permafrost ecosystem" This paper is interesting and relevant to the communities. There are only few dataset from these continuous permafrost systems, so these results are extremely valuable. And there are even fewer that can span a long temporal range and compare the short term to the long term impact of drainage on the fluxes. However, while this idea is definitively relevant, the sparse nature of the data (2002-2004, and 2013-2015) is a bit too short to be able to really say something about a decadal change, especially considering the large interannual variability in the fluxes of these systems. I suggest the authors to be careful about overstepping the boundary of what they can say with their data. On the other hand the vegetation change is definitively something that can be safely highlighted, and potentially better quantified using remote sensing data? The signature of shrubs is very different than cotton grasses, so high resolution images should allow quantifying the vegetation change due to drainage. Specific comments Page 1 lines 18-19: not clear, this statement seems in contradiction with the previous one saying that "$CO_2$ uptake to be systematically reduced within the drained area", rephrase highlighting that there was a recover? Page 1 line 24- : this is an interesting idea, unfortunately there are not enough data to really do a trend analysis, three years (2002-2004., which looking at Fig. 2 is really two summers and a half) are compared to another three (2013-2015, again as shown in Fig. 2 two summers and a half), and given the large interannual variability in fluxes of these systems, these a little too few to say something about a decadal change. MODIC is mentioned here, but it is not indicated how it was used to support this argument... Page 4 line 105: is this disturbed site the same as the site operated in 2002-2005? Page 6 line 187-190: what is this definition based on? Any reference or underlying reason to support this choice? Also, it is not clear how exactly the other periods were defined, it is only mentioned what was the start of the pre-season and the end of season, what about the other periods? Please describe in more careful details how the beginning and end of each period were defined. Their length also seem to be different depending on the year. It is relevant to understand how these dates changed among these three years and why. Page 7 lines 205-206: is this 19 gC from the drained or the control site? It should be mention the loss in both of them. Also, how much did the loss vary among these three years? Include st. deviations to the estimate for each site. Also, in 2014 the loss from the drainage site is very similar to the reference site, any reason for this? Figure 4: the colors seem a bit off, a standardization would help to compare these pictures, maybe using the grey element (a boat?) in the right of the picture? This grey should be used to balance the colors. Page 8 line 231:"both" "both" repetition...remove one Page 14 line 425: are really the gaps evenly distributed? I always saw much higher data coverage during summer and less early and later in the season...

---

## Author Comment (AC1) · 20 Jul 2016

We thank Anonymous Referee #1 for the constructive feedback and the helpful comments/suggestions that helped improving the manuscript. Below comments from the referee followed by our responses can be found.

Line 16: clarify if you mean 'net $CO_2$ uptake' or GPP Net $CO_2$ uptake was meant here, and it was clarified in the manuscript by using the term sink strength.

Line 22: it wasn't clear until I read the paper, what you meant by 'intensified', may want to clarify There was a sub sentence added to describe 'intensified'.

Line 30: I suggest updating to Hugelius (2014) and updated permafrost C numbers

[Figure]

The citation was updated as suggested.

Line 55: I don't think you mean methanotrophic here, which can occur under aerobic and anaerobic conditions. Change to methanogenic, if that's what you meant. We used the term 'microbial decomposition of methane' instead.

Line 60: There are quite a few long-term studies from Toolik lake e.g., Mack et al (2004), Sistla et al (2013) The study from Mack et al (2004) was integrated. We refer to Sistla et al (2013) in more detail a few sentences later, and skipped it here to avoid revisions.

Line 116: I'm confused by 'during the first nine months' since the measurements only occur during the growing season. To avoid misunderstandings we corrected the statements since open-path data were used during the first growing season only.

Line 128: change 'data are collected' to data 'were'. Throughout methods, change to past tense. We changed all sentences to past tense in Section 2 of the manuscript.

Methods: Are missing a lot of information about gapfilling and partitioning the fluxes. But then I see this information is in the Discussion, which is a distraction from the discussion section and gap in the Methods. Most of that text in Section 4.3 should be moved to the Methods, the remaining could go in the Supplement. We rearranged parts of the text paragraphs as the reviewer suggested. More information about the gapfilling and flux partitioning method can be found in Section 2.4 now. Furthermore the remaining text of Section 4.3 is used as supplement material.

Line 160: Change 'presented' to 'present' It was corrected as the reviewer suggested.

Line 180: not sure what is meant by 'missing release' It was clarified in the section. We referred to the fact that the expected release with dominating positive fluxes at the beginning of the growing season after the snowmelt is completely missing in these time series.

Methods: Add in information about statistical design and analysis. We added some

information on the statistical analysis covering the ANOVA as a new paragraph at the end of section 2.4.

Line 207: I suggest to change 'evolving' to 'greening' or 'changing' It was changed to 'greening' as the reviewer suggested.

Line 215: change 'higher cumulative' to 'higher net cumulative' It was corrected as the reviewer suggested.

Line 255-266: Given that there was only one historical year of data from the disturbed site, and given the level of inter-annual variability, I don't think the data support such a strong statement. I would change 'demonstrates' to 'suggests' and 'strongly rebounded' to 'may have rebounded' It was corrected as the reviewer suggested.

Figures 7–9: Would be easier to see trends if you changed the line style for either the decadal or recent data so that one set is solid line and the other is dashed. The style of the graphs was changed as suggested with dashed lines for the historic dataset.

Line 190: ANOVA not mentioned in methods; as noted above, need to add in a statistical analysis section to the methods See comment for Methods above.

Line 306-307: How was the effect of soil temperature, moisture, etc. analyzed? Results presented in companion studies (Kwon et al, BGD 2016; Goeckede et al., in prep.), based on small scale flux observations (flux chambers) and associated observations of environmental conditions; clearly demonstrate the influence of soil moisture on soil temperature conditions. Here, drier soils promoted warmer conditions near the soil surface, and colder conditions at deeper soil layers. The warmer top soils promoted microbial respiration of recently assimilated carbon (Kwon et al., BGD 2016). With the focus on eddy-covariance observations in the present manuscript, such relationships are harder to establish, since the eddy covariance footprint always covers a mixture of wet and dry microsites, respectively, and observations of soil temperature/moisture are only available for very few spots. Therefore, we added a link to Kwon et al. (2016) as a

reference for more process-based analyses of the carbon cycle at this study site.

Line 317: manuscript doesn't present organic matter decompostion results. Change to 'decomposition of soil organic matter' to 'ecosystem respiration' It was corrected as the reviewer suggested.

Line 323-33: The APEX peatland drainage experiment (Alaska) has been running since ~2005. Some of these pubs aren't long-term but they are water drainage studies in the permafrost regions, and their results may inform your discussion: Turetsky (2008, APEX), McConnell et al (2013, APEX), Natali et al (2015, not APEX) We decided not to include the proposed literature since none of the studies is matching our targets. Either the focus is on methane in combination with the drainage experiment, the measurement technique is different or the manipulation is focusing on warming, which is in our case a secondary effect of the drainage but not the primary disturbance. To make clear what focus area we are targeting, we changed 'carbon fluxes' to the more specific term 'summertime CO2 fluxes'.

Line 343: change 'annual' to 'growing season' It was corrected as the reviewer suggested.

Line 378-379Any long-term changes in snow-depth? Or winter soil temps? Measurements were carried out during the growing season only. No data on soil temperature or snow depth were collected during the off-seasons, e.g. winter. Therefore no information can be given for this period. BEST-data indicate that wintertime air temperatures have not changed significantly over the last decade but can be highly variably from year to year.

Line 485: What do you mean exactly by adaptations? The sentence was restructured to avoid misunderstandings.

---

## Author Comment (AC2) · 20 Jul 2016

We thank Anonymous Referee #2 for the constructive feedback and the helpful comments/suggestions that helped improving the manuscript. In the sections below, comments from the referee are followed by our responses.

This paper is interesting and relevant to the communities. There are only few dataset from these continuous permafrost systems, so these results are extremely valuable. And there are even fewer that can span a long temporal range and compare the short term to the long term impact of drainage on the fluxes. However, while this idea is definitively relevant, the sparse nature of the data (2002-2004, and 2013-2015) is a bit too short to be able to really say something about a decadal change, especially

[Figure]

considering the large interannual variability in the fluxes of these systems. I suggest the authors to be careful about overstepping the boundary of what they can say with their data. On the other hand the vegetation change is definitely something that can be safely highlighted, and potentially better quantified using remote sensing data? The signature of shrubs is very different than cotton grasses, so high resolution images should allow quantifying the vegetation change due to drainage.

The authors agree with the reviewer that observational datasets of surface-atmosphere exchange processes are often subject to pronounced interannual variability, and that long-term continuous observations are necessary to allow the identification of statistically significant trends. Accordingly, our two observation periods, i.e. the 'historic' data from 2002-05, and the 'recent' data from 2013-15, are too short to claim that each of them fully represents the average conditions within each of these timeframes; however, given the fact that we observed characteristic patterns (e.g. the maximum $CO_2$ uptake rate in summer) where conditions could clearly be separated between historic and recent periods, resp., we believe that even this limited database can provide solid evidence that the ecosystem has undergone systematic shifts over the past ∼12 years. We highlighted the issue of data representativeness, and uncertainty induced by interannual variability, by adding new text to the discussion section of this manuscript. Regarding the analysis vegetation change over time based on remote sensing data, long-term trends can certainly be worked out, but an in-depth analysis is complicated by the fine-scale variability of ecosystem characteristics, and year-to-year variability in phenology timing. A quantitative analysis that goes beyond the description of trends in phenology therefore is beyond the scope of the presented manuscript, and is planned to be covered in a follow-up study that has a stronger focus on remote sensing methods.

Specific comments

Page 1 lines 18-19: not clear, this statement seems in contradiction with the previous one saying that "CO2 uptake to be systematically reduced within the drained area",

rephrase highlighting that there was a recover? The sentence was restructured to avoid misunderstandings.

Page 1 line 24- : this is an interesting idea, unfortunately there are not enough data to really do a trend analysis, three years (2002-2004., which looking at Fig. 2 is really two summers and a half) are compared to another three (2013-2015, again as shown in Fig. 2 two summers and a half), and given the large interannual variability in fluxes of these systems, these a little too few to say something about a decadal change. MODIC is mentioned here, but it is not indicated how it was used to support this argument.

As already outlined above, we agree with the reviewer that the analysis would be stronger with higher data coverage. Since most of our analyses that target long-term shifts in flux patterns are based on qualitative evaluation of trends, and an ANOVA was applied to the peak NEE uptake only, we changed the text accordingly to avoid misunderstandings. The MODIS-based classification was applied to remove the influence of year-to-year shifts in phenology, e.g. linked to the timing of snow meld, and thus facilitate a comparison between data years. Our results demonstrate that this normalization removes a large part of the interannual variability in cumulative flux budgets, and makes possible the separation of data years into two distinct groups, the recent (2013–2015) and the historic (2002–2004) datasets.

Page 4 line 105: is this disturbed site the same as the site operated in 2002-2005? During the 'historic' observation period (2002-05), one single tower was used to monitor fluxes for control (2002-04) and drained (2005) conditions. The exact same tower (which was still in place ∼10 years later) was chosen to install the new eddy covariance instrumentation for the drainage (disturbed) tower. This fact is now clearly indicated by a new sub-sentence in Section 2.5, which describes the 'historic' experiment setup.

Page 6 line 187-190: what is this definition based on? Any reference or underlying reason to support this choice? Also, it is not clear how exactly the other periods were defined, it is only mentioned what was the start of the pre-season and the end of

season, what about the other periods? Please describe in more careful details how the beginning and end of each period were defined. Their length also seem to be different depending on the year. It is relevant to understand how these dates changed among these three years and why.

The definition of our so-called 'key dates' is based on the analysis of temporal trends in MODIS time series of NDVI. The detailed method will be presented in a companion manuscript that is currently in preparation. In the revised version of this manuscript, we added information on how key dates were derived.

Page 7 lines 205-206: is this 19 gC from the drained or the control site? It should be mention the loss in both of them. Also, how much did the loss vary among these three years? Include st. deviations to the estimate for each site. Also, in 2014 the loss from the drainage site is very similar to the reference site, any reason for this?

Measurements in 2013 started mid-July, therefore only two years (2014-2015) can be analyzed for this sub-season. Adding individual standard deviations by treatment might not be useful since only two time series can be used for this analysis. Accordingly we decided to merge pre-season data across treatments (drained and references) and years (2014-2015), resulting in an average release of 19 gC m-2 during this sub-season. This was clarified in the text. Differences between drained and reference site are lower in 2014 due to smaller differences in soil temperatures.

Figure 4: the colors seem a bit off, a standardization would help to compare these pictures, maybe using the grey element (a boat?) in the right of the picture? This grey should be used to balance the colors. The colors were corrected as suggested.

Page 8 line 231:"both" "both" repetition remove one. It was corrected as the reviewer suggested.

Page 14 line 425: are really the gaps evenly distributed? I always saw much higher data coverage during summer and less early and later in the season. There certainly is

year-to-year variability, and also differences in data gaps exist between towers, but the data coverage between pre-, early-, late- and post-season did not differ systematically.

---

## Author Response (AR2)

**Author response to comments of associate editor**

We thank associate editor for the constructive feedback and the helpful comments/suggestions that helped improving the manuscript. Below comments from the associate editor in black followed by our responses marked in blue can be found.

1- You wrote at line 25: "Net changes in CO2 exchange fluxes are dominated by a major increase in photosynthetic uptake, resulting in a stronger CO2 sink in 2013–2015."
I think that the expression "CO2 exchange fluxes" is redundant (although I am not native English speaker). I think you need to choose one or the other between "CO2 exchange" or "CO2 fluxes".
"Exchange" was deleted.

2- You wrote at line 167: "Longer gaps were caused by flooding events, when the entire system needed to be shut down to avoid damage, and for a single event as a result of an additional laser offset in the closed-path gas analyzer at Tower 1 in 2015."
I think the second part of sentence (after comma) needs to be revised maybe as follows: "or resulting from a laser offset in the closed-path gas analyzer at Tower 1 in 2015".
It was corrected as the associate editor suggested.

3- What is this "previously" doing there at line 176?
"Previously" was deleted.

4- You wrote at line 177: "This decision was based on our observance of implausible patterns in near-surface temperature measurements at our site, and as, because of a damped diurnal cycle, higher-quality soil temperatures at 0.08 m below the surface cannot be correlated to short-term changes in respiration patterns."
This sentence needs revision. I suggest the following but anything else is fine: "This decision was based on two factors: first, the patterns in near-surface temperatures were implausible at our site, and secondly, higher-quality soil temperatures at 0.08 m below the surface cannot be correlated to short-term changes in respiration patterns because of a damped diurnal cycle."
It was corrected as the associate editor suggested.

5- At line 184, I suggest: "For the statistical analysis of the whole dataset, a linear regression with a moving window of 14 days was used to analyze the peak uptake. An hourly shift was chosen to account for short-term variabilities."
It was corrected as the associate editor suggested.

6- I suggest the following changes in your sentence of line 188: "Measurements and thus errors are independent, the data are normally distributed and the variances are equal in most cases."
It was corrected as the associate editor suggested.

7- I suggest a few changes in the paragraph starting at line 380: "Here, we compare two observation periods, the historic (2002-05) and the recent dataset (2013-15), both covering three years of reference measurements. These timeframes might not be sufficient to fully represent conditions within each observation period, since surface-atmosphere exchange processes are often subject to pronounced interannual variability. Long-term continuous observations would allow a more in-depth statistical analysis. However, we observed characteristic patterns (e.g. the maximum CO2 uptake rate in summer) that clearly showed a distinctive trend between historic and recent periods. Thus, even from this limited database, we can see that the ecosystem has undergone a systematic shift over the past ~12 years."
It was corrected as the associate editor suggested.

8- At line 465, I suggest: "CO2 flux data from recent years indicate that the drained ecosystem has adapted to the disturbance, with changes in the biotic and abiotic conditions leading to a rebound to previous carbon flux levels."

It was corrected as the associate editor suggested.

9- At line 710, maybe replace executed by computed or obtained?
We replaced it by computed.

[revised manuscript text omitted]